# Preparation of an Antioxidant Assembly Based on a Copolymacrolactone Structure and Erythritol following an Eco-Friendly Strategy

**DOI:** 10.3390/antiox11122471

**Published:** 2022-12-15

**Authors:** Aurica P. Chiriac, Alina Ghilan, Alexandru-Mihail Serban, Ana-Maria Macsim, Alexandra Bargan, Florica Doroftei, Vlad Mihai Chiriac, Loredana Elena Nita, Alina Gabriela Rusu, Andreea-Isabela Sandu

**Affiliations:** 1Petru Poni Institute of Macromolecular Chemistry, 41 A Grigore Ghica Voda Alley, 700487 Iasi, Romania; 2Faculty of Electronics Telecommunications and Information Technology, Gh. Asachi Technical University, Bd. Carol I no. 11A, 700506 Iasi, Romania

**Keywords:** bio-based copolymacrolactone, ethylene brassylate, squaric acid, suspension copolymerisation, erythritol, biocompatibility and antioxidant behaviour

## Abstract

The study presents the achievement of a new assembly with antioxidant behaviour based on a copolymacrolactone structure that encapsulates erythritol (Eryt). Poly(ethylene brassylate-co-squaric acid) (PEBSA) was synthesised in environmentally friendly conditions, respectively, through a process in suspension in water by opening the cycle of ethylene brassylate macrolactone, followed by condensation with squaric acid. The compound synthesised in suspension was characterised by comparison with the polymer obtained by polymerisation in solution. The investigations revealed that, with the exception of the molecular masses, the compounds generated by the two synthetic procedures present similar properties, including good thermal stability, with a T_peak_ of 456 °C, and the capacity for network formation. In addition, the investigation by dynamic light scattering techniques evidenced a mean diameter for PEBSA particles of around 596 nm and a zeta potential of −25 mV, which attests to their stability. The bio-based copolymacrolactone was used as a matrix for erythritol encapsulation. The new PEBSA–Eryt compound presented an increased sorption/desorption process, compared with the PEBSA matrix, and a crystalline morphology confirmed by X-ray diffraction analysis. The bioactive compound was also characterised in terms of its biocompatibility and antioxidant behaviour.

## 1. Introduction

In recent years, monomers from renewable sources have been increasingly used in the synthesis of polymeric materials due to their biodegradability potential, as well as the abundant sources of raw materials derived from fossil fuels. Macrolactones, bio-based compounds obtained from renewable resources and belonging to this category of monomers, ensure the synthesis of polyesters, which, due to their various and significant properties, such as biocompatibility, biodegradability, and non-toxicity, find different end-uses, including biomedical applications [1].

The most widely used method of obtaining polyesters from macrolactone is ring-opening polymerisation (ROP) performed catalytically or enzymatically, in bulk, solution, emulsion, or suspension. At the same time, the differences should be noted between the conditions of ring-opening polymerisation between small lactones and macrolactones that are caused by little or no ring strain in the case of the compounds with 12 or more atoms, which requires specific catalysts to obtain optimal yields [2,3,4,5,6,7,8]. Recently, one of the most studied macrolactones that is of particular interest is ethylene brassylate (EB), a 17-membered lactone ring, investigated essentially by ring-opening polymerisation [9,10,11,12,13,14,15,16,17,18,19]. The synthesised homopolymer, as well as other different EB-based block/copolymers, can provide an amphiphilic character; for example, in the poly(ethylene glycol)-b-poly(ethylene brassylate) copolymer [10], a higher crystallisation rate of about 67 °C and chains easily susceptible to hydrolytic degradation in the case of copolymers based on D, L-lactide and EB [13] improved mechanical behaviour, flexibility, and ductility compared to polylactides, since only a molar content of 5% EB comonomer increases the elongation at break from 2 to 87% [13], and also due to a self-assembled morphology, building block capacity, and network formation. In one of the mentioned references [9], the ethylene brassylate homopolymer was synthesised in bulk or solution with benzyl alcohol and various organic compounds as initiation systems, and the study highlighted the interdependence between the molecular mass of the polymer and the catalysts used. Our group investigated the polymerisation of EB with squaric acid (SA) at various rates, and the newly synthesised co-polymacrolactone systems presented increased functionality that opened new opportunities for the application of these compounds [20,21]. SA, a strong acid due to its polar units, is capable of chemical processes, can form intermolecular hydrogen bonds, and, furthermore, can develop polymeric networks with various applications, including medicinal chemistry [22]. Poly(ethylene brassylate-co-squaric acid) (PEBSA) was synthesised in a homogeneous system in dioxane and using 1-hexanol as a catalyst. The study proved that PEBSA has special properties, such as good thermal stability (T_onset_ starting from 259 °C), which increases with the SA content in the PEBSA composition, thermo-responsive behaviour, higher value of the dielectric constant, and an increased potential for supramolecular self-organised structures, while the presence of polar functional groups available at the surface confers an affinity towards hydrophobic substances. Encouraging data on the characteristics of PEBSA led us to continue our research to synthesise the copolymacrolactone in eco-friendly conditions and to find new potential applications. Moreover, there has been significant interest in developing synthetic ways to prepare this type of material with the reduced use of toxic products (reactant, catalysts, and solvents, for example) and without the need for tedious purification methods [1,23].

Thus, the study was conducted to obtain’PEBSA in aqueous suspension. Suspension polymerisation has several advantages over other polymerisation techniques. These include the mild reaction conditions, the continuous phase is usually water, which acts as a very efficient heat transfer medium (in addition, it is very economical and more environmentally friendly than organic solvents), and the separation of the obtained compounds is much easier than in solution polymerisation [24]. 

Complex systems that benefit from the functional contribution of the constituent elements make them much more competitive. The use of PEBSA as a biopolymer-based functional structure in a new chemical system also contributes to the diversification of its range of uses. In addition, by using PEBSA, which has already demonstrated its biocompatibility in vivo [21], in creating a new structure by encapsulating a suitable specific compound, its versatility is highlighted again. 

Erythritol (Eryt), which belongs to the polyol family and has a variety of biological functions, is generally used for the substitution of sugar, as studies have already shown the absence of adverse effects associated with its use [25]. Moreover, recent findings have revealed that it can actively support the maintenance of both oral and systemic health. Eryt effectively inhibits the formation of dental plaque, lowers levels of acetic acid in plaque, reduces the number of *Streptococcus mutans*, *S. gordonii*, and *Porphyromonas gingivalis* in saliva, and decrease the overall incidence of dental caries [26]. Emerging evidence also shows that Eryt has antidiabetic or antihyperglycemic effects, exerts antioxidative properties by efficiently scavenging hydroxyl radicals (OH*), and displays an endothelium-protective effect [27,28].

In this context, the study presents a more attractive alternative for the synthesis of the PEBSA copolymacrolactone system through an environmentally friendly approach that does not contain harmful organic solvents. By designing the synthesis of PEBSA in suspension, in the presence of a suitable dispersant, we aimed to obtain the new bio-based copolymacrolactone not only in eco-friendly conditions but also with improved characteristics. The comparative characterisation of the copolymacrolactone synthesised by the two processes, in suspension and solution, was carried out, followed by the evaluation of the polymer structure as an Eryt coupling matrix, with the aim of obtaining a new bioactive compound with antioxidant properties. Thus, the research provides information on the preparation of an efficient and practical biocompatible and antioxidant structure with properties and potential for further development.

## 2. Materials and Methods

### 2.1. Materials 

All chemicals were used as received without further purification. Ethylene brassylate (EB, 1,4-Dioxacycloheptadecane-5,17-dione, C_15_H_26_O_4_, M_w_ = 270.36 g/mol, purity (GC) > 95.0%), squaric acid (SA, 3,4-dihydroxy-3-cyclobutene-1,2-dione, H_2_C_4_O_4_, M_w_ = 114.06 g/mol, purity (HPLC) > 99.0%), p-dodecylbenzene sulphonic acid sodium salt (DBSNa), meso-erythritol (Eryt), and dioxane were purchased from Sigma-Aldrich Co. (Burlington, MA, USA), while 1-hexanol anhydrous was acquired from Across-Organics. The ultrapure water used in all the synthesis was prepared with a Milli-Q device.

### 2.2. Preparation of Samples 

#### 2.2.1. Synthesis of the Copolymacrolactone 

Two variants of PEBSA copolymacrolactone were synthesised. The copolymer variant obtained in solution and coded PEBSA_solution was synthesised as previously presented [2,20,21]. Briefly, the ring opening of EB and polycondensation with SA (comonomers in a molar ratio of 75/25 EB/SA) took place in a homogeneous system of 1, 4 dioxane, and using 1-hexanol as an initiator (in 10/1 ratio related to EB content). The reaction was conducted in a vessel immersed in a temperature-controlled oil bath at 100 °C, under nitrogen atmosphere, with a stirring rate of 250 rpm. Predetermined amounts, namely, 0.0075 moli = 1.946 mL EB, 0.0025 moli = 0.285 g SA, and 10 mL 1,4 dioxane were allowed to react for 24 h. The PEBSA_suspension variant additionally contains DBSNa as a dispersant (in 10/1 ratio related to EB content). The reaction was carried out in deionised water at 90 °C, under nitrogen atmosphere, for 24 h, and with a stirring rate of 250 rpm. PEBSA_solution was precipitated in water, while PEBSA_suspension was separated by centrifugation. Both samples were washed with water repeatedly, then lyophilised and stored in the refrigerator for further analysis and testing.

#### 2.2.2. Preparation of the Bioactive Compound 

A mixing method was used to prepare the bioactive compound. Briefly, the PEBSA_suspension variant was solubilised in dioxane and mechanically stirred for 24 h to prepare a 5% solution. Then, a 5 wt % solution of Eryt in water was added to the copolymer solution in a 1:1 ratio and gently stirred for about 24 h, while PEBSA–Eryt bioactive compound nanoparticles are formed. Further, to analyse their characteristics, the PEBSA–Eryt particles were allowed to stand for 24 h, centrifuged, and then dried. In the subsequent analyses and characterisations, we considered that the entire amount of erythritol used for the formation of the bioactive compound was fully encapsulated in the polymacrolactone network.

### 2.3. Sample Characterisation 

#### 2.3.1. Spectroscopic Measurements 

##### FTIR Spectroscopic Analyses

FTIR spectra of copolymers and bioactive compound were recorded on a Vertex Brucker Spectrometer in an absorption mode ranging from 400 cm^−1^ to 4000 cm^−1^. Dried samples were mixed with potassium bromide IR grade, ground, and compressed into a disc shape before recording. The spectra were acquired at 4 cm^−1^ resolution as an average of 64 scans and are presented in Figure 1.

##### ^1^H-NMR Spectroscopic Analyses 

NMR spectra were recorded on a Bruker Neo Instrument (Bruker BioSpin, Rheinstetten, Germany) operating at 400.1 MHz for ^1^H. Samples recorded with a 5 mm multinuclear inverse detection z-gradient probe at room temperature in DMSO-d6 present the chemical shifts reported in δ units (ppm) for ^1^H chemical shifts referenced at 2.51 ppm and with assignments presented in Figure 2. 

**^1^H NMR** (400.1 MHz, DMSO-d6, δ, ppm) **PEBSA_solution**: 11.29 (bs, 0.43H, OH), 4.00 (t, 2.7H, J = 6.6 Hz, CH_2_-O, a), 2.19 (t, 14.5H, J = 7.4 Hz, -O-CH_2_-C=O, b), 1.54 (m, 19.3H, CH_2_-c),1.27 (s, 65.4H, CH_2_-d), 0.87 (t, 3H, J = 6.9 Hz, CH_3_-f).

**^1^H NMR** (400.1 MHz, DMSO-d6, δ, ppm) **PEBSA_suspension**: 11.96 (s, 13.2H, OH), 4.00 (t, 1.94 H, J = 6.13 Hz, **a**), 2.19 (t, 28.4H, J = 7.10 Hz,**b**), 1.49 (m, 35.1Hz, **c**), 1.25 (s, 119.5 Hz, **d**), 0.87 (t, 3H, J = 6.6 Hz, , **f**).

**^1^H NMR** (400.1 MHz, DMSO-d6, δ, ppm) **PEBSA_Eryt**: 11.95 (s, 11.9H, OH), 4.44–4.21 (m, 63.11H OH-erythritol), 4.00 (t, 2.8H, J = 6.54 Hz, **a**), 3.55 (m, 37.7H-eritritol CH_2_), 3.37 (s, 22.74H + 76.74H, eritritol-CH+CH_2_+H_2_O), 2.19 (t, 30.7H , J = 7.34 Hz, , **b**), 1.49 (m, 36.34H, **c**), 1.25 (s, 127.11H, **d**), 0.87 (t, 3H, J = 7.16 Hz, **f**).

#### 2.3.2. Molecular Weights of the Copolymacrolactones 

Comparative molecular weights of PEBSA_suspension and PEBSA_solution samples, determined by Gel Permeation Chromatography (WGE SEC-3010 multi-detection system, with two PL gel columns, a dual detector RI/VI calibrated with PS standards (580-1,350,000 DA), and a chloroform flow rate of 1.0 mL/min at 30 °C), are presented in Table 1. The molecular weights data, analysed by using PARSEC Chromatography software, present a measurement error of up to 3%.

#### 2.3.3. Thermogravimetric Analysis

The thermal behaviour of the copolymacrolactone samples was studied using an STA 449 F1 Jupiter thermo-balance made by Bruker Germany. The dried samples weighing from 10 to 12 mg were placed in an open Al_2_O_3_ crucible and heated in dynamic mode from room temperature up to 675 °C. Runs were performed with a heating rate of 10 °C/min in a nitrogen atmosphere with a gas flow of 40 mL/min. Differential thermal analysis was recorded simultaneously on the same apparatus using Al_2_O_3_ as reference material. Data collection and processing were performed using Proteus 5.0.1 software (Proteus, Visalia, CA, USA). 

#### 2.3.4. Size Measurements

Size determination of the PEBSA_suspension and PEBSA_Eryt samples was made with the dynamic light scattering technique using a Zetasizer model Nano ZS device, with red laser 633 nm He/Ne from Malvern Instruments, Malvern, UK. The system uses a non-invasive back scatter (NIBS) technology (which reduces the multiple scattering effects), wherein the optics are not in contact with the sample, back scattered light being detected. During determinations, the Mie method was applied over the whole measuring range from 0.6 nm to 6 μm. Analyses were performed on a 2 mL sample of PEBSA_suspension (dispersed in dioxane), Eryt (dispersed in water), and PEBSA–Eryt compounds, as obtained after mixing the initial constituents in appropriate concentrations of 5%. Zeta Potential (ZP) was also determined by using the Smoluchowski relationship on samples dispersed in water with a concentration of 5%.

#### 2.3.5. Dynamic Vapours Sorption Measurements

The behaviour of the samples in the presence of moisture was studied by determining their water vapour sorption capacity in a dynamic regime using a fully automated gravimetric device, IGAsorp, made by Hiden Analytical (Warrington, UK). The most important part of this equipment is an ultrasensitive microbalance which measures the weight change as the humidity is modified in the sample chamber at a constant temperature. The measurements are controlled by a user-friendly software package. After the samples were placed in a special container, they were dried at 25 °C in flowing nitrogen (250 mL/min) until their weights were in equilibrium at a relative humidity (RH) of less than 1%. Then, the RH was gradually increased from 0 to 90%, in 10% humidity steps, each having a pre-established equilibrium time between 10 and 20 min, and the sorption equilibrium was obtained for each step. The RH was decreased and the desorption curves were registered [29].

#### 2.3.6. X-ray Diffraction Analysis

X-ray diffraction analysis was performed on a Rigaku Miniflex 600 diffractometer using CuKα-emission in the angular range of 2–50º (2θ) with a scanning step of 0.0025º and a recording rate of 1º/min.

#### 2.3.7. Morphological Characterisation

Scanning electron microscopy (SEM) studies were performed on PEBSA samples fixed in advance by means of colloidal copper supports. Their morphology was examined with a Quanta 200 Scanning Electron Microscope (FEI) operating at 20 kV in Low Vacuum mode using a secondary electron detect

or LFD. Transmission electron microscopy (TEM) micrographs were obtained by using a Hitachi HT7700 transmission electron microscope operating at 100 kV. 

#### 2.3.8. Antioxidant Behaviour of PEBSA–Eryt Bioactive Compound

The antioxidant activity of the compound was evaluated by using a stable free radical known as 2,2-diphenyl-1-picrylhydrazyl (DPPH), according to the literature with some modifications. The experiment was performed according to the methodology described by Brand-Williams et al. [30,31]. Briefly, the samples (PEBSA_Eryt and Eryt) were reacted with the stable DPPH radical in an ethanol solution. The reaction mixture consisted of adding 0.05 g of the sample in 3 mL of ethanol and 0.3 mL of DPPH radical solution in 0.5 mM of ethanol. The reduction in DPPH radicals was determined by measuring the absorption at 517 nm after predetermined periods of time by using a UV–Vis SPECORD 200 Analytik Jena spectrophotometer. The mixture of ethanol (3.3 mL) and sample (0.05 g) serves as blank. The control solution was prepared by mixing ethanol (3 mL) and DPPH radical solution (0.3 mL). The scavenging activity percentage (AA%) was determined according to the following equation [32]:(1)$$\mathrm{AA}\%=100-\left[\frac{(Abs_{sample}-Abs_{blank\;})\times 100}{Abs_{control}}\;\right]$$

#### 2.3.9. Biocompatibility of PEBSA–Eryt Bioactive Compound

The materials used in this assay were: HGF (human gingival fibroblast) cells purchased from CLS (Eppelheim, Germany), Eagle’s Minimal Essential Medium alpha (aMEM) and antibiotic-antimycotic mixture from Lonza (Verviers, Belgium), foetal bovine serum (non-USA origins) from Sigma Aldrich (Schnelldorf, Germany), TrypLETM Express Enzyme from Gibco (Langley, VA, USA), phosphate-buffered saline (PBS) from Invitrogen (Eugene, OR, USA), CellTiter 96^®^ AQueous One Solution Cell Proliferation Assay (MTS assay) from Promega (Madison, WI, USA), and CytoOne^®^ 96-well plates from StarLab (Hamburg, Germany). The absorbance was measured with a FLUOstar Omega Filter-based multi-mode microplate reader from BMG LABTECH (Offenburg, Germany). 

Normal human gingival fibroblasts (HGF cells) were cultivated in cell culture flasks with complete Eagle’s Minimal Essential Medium alpha (aMEM) containing 1% antibiotic-antimycotic mixture and 10% foetal bovine serum under 5% CO_2_ humidified atmosphere at 37 °C. TrypLETM Express Enzyme was used for detaching the cells. Stock solutions for treatment (0.2%) were prepared with DMSO and working solutions (0.0002%) were obtained by diluting the stock solutions with complete medium, so that the final concentration of DMSO in the cell culture was 0.1%. Working solutions were used for incubation. Control cells were treated only with complete cell culture medium containing 0.1% DMSO.

The biocompatibility was determined by evaluation of the relative viability of normal human gingival fibroblasts (HGFs) in contact with the sample solutions, using MTS assay. Therefore, HGF cells were plated in 96-well plates at a density of 5 × 10^3^ cells/well in 100 μL aMEM medium/well and incubated for 24 h. After 24 h, the culture medium was replaced with working solutions and the plates were incubated for another 24 h. Next, 20 μL MTS solution/well were added and the plates were incubated for 1–4 h. Finally, the absorbance at λ = 490 nm was measured with a microplate reader. The relative cell viability is expressed as a percentage of the viability of control cells.

Data analysis was performed with GraphPad Prism software version 7.00 for Windows (GraphPad Software, San Diego, CA, USA). The obtained results represent the mean ± standard error of the mean (S.E.M.). The statistical significance between groups was determined by one-way ANOVA with Tukey’s multiple comparisons test and values of *p* < 0.05 were considered significant.

### 2.4. Statistical Analysis 

The data presented in this study are the average of triplicate experiments and have the standard error of the mean (S.E.M.). In addition, the statistical differences between data were made by one-Way ANOVA with Tukey’s test for the presented results. One-Way ANOVA with Tukey’s test and the bivariate Pearson Correlation were also used for the statistical differences between samples that resulted for the antimicrobial and biocompatibility tests of the samples.

## 3. Results and Discussion

### 3.1. FTIR Spectra

The similarities of the two spectra resulting from FTIR spectroscopy analyses (Figure 1a) confirm that both copolymer variants obtained by alternative synthesis methods have the same chemical structure. In the Supporting Information (Appendix A), the most important registered frequencies and their spectral assignments are presented. Furthermore, the FTIR spectra corresponding to the bioactive complex and the constituent components highlight the differences that occurred (Figure 1b) after the formation of the compound. Thus, Eryt has a distinctive O-H stretch in the range of 3200 to 3500 cm^−1^, as well as strong absorptions at 1054 and 1075 cm^−1^ due to the stretching vibrations of C–O group and at 964 cm^−1^ due to the rocking vibrations of CH_2_. The PEBSA spectra illustrate, in agreement with the literature data, characteristic peaks of the two monomers, EB and SA, but also changes that confirm the synthesis of the copolymacrolactone [20]. This includes the opening of the macrolactone cycle and the polycondensation process justified by the disappearance of OH peaks characteristic of squaric acid. Therefore, the most representative bands in the PEBSA_suspension spectra are the asymmetric and symmetric vibrations of -CH_2_ bonds that occur at 2923 cm^−1^ and 2852 cm^−1^ and the carbonyl (C=O) symmetric stretching vibration registered at 1695 cm^−1^. Other important spectral features appeared at 1461 cm^−1^ associated to C-H bending, and at 1280 and 1228 cm^−1^ representative of C-O stretching. The band at 1189 cm^−1^ is associated with C–O–C bending. FTIR-based comparative analysis also confirms the formation of the bioactive compound. Therefore, bands corresponding to both components can be found in the PEBSA–Eryt spectrum. Moreover, as shown in Figure 1b, several modifications in the absorption bands are observed, especially in the case of the bands assigned to the OH stretch in the region 3250–3400 cm^−1^. The splitting of the band, as well as a shift to slightly lower frequencies, demonstrates the appearance of intermolecular bonds between the synthesised copolymacrolactone and the polyol. The OH band of the associated species absorbs at lower wavenumbers, as the hydrogen bonding weakens the OH band, thus showing an increase in the interactions between PEBSA_suspension and Eryt [9,33]. No new chemical peak appeared in the spectrum of PEBSA–Eryt apart from those of the used compounds, which confirms the physical nature of interactions between the constituent elements in the bioactive structure.

### 3.2. H-NMR Spectra 

By comparing the ^1^H-NMR of the two copolymers it can be seen that PEBSA_suspension sample has a double degree of polymerisation compared to PEBSA_solution (Figure 3), which is in good agreement with the molecular weight resulting from GPC determinations. The PEBSA–Eryt spectrum confirms the presence of Eryt in the system, as a physical mixture with the copolymacrolactone, by the signals from 4.35–4.21 ppm (OH), 3.55 ppm, and 3.37 ppm (CH and CH_2_). In addition, from the ^1^H-NMR spectra overlap, it is observed that the PEBSA_suspension and PEBSA_Eryt samples are close in molecular mass/degree of polymerisation, and there is no chemical interaction between PEBSA and erythritol.

Additional information regarding the structural analysis of the realised systems are presented in the Supporting Information (Appendix A) of the article.

### 3.3. Thermal Degradation

The thermal behaviour of the PEBSA samples was investigated by simultaneous thermo-gravimetric and differential thermal analysis. Both analyses provide valuable information about the thermal stability and decomposition pathway and illustrate how the thermal properties of the samples are affected in relation to their synthesis procedures. Figure 4a displays the curves for weight loss and differential thermal analysis in relation to temperature, whereas Figure 4b displays the curve for weight loss’s first derivative in relation to temperature. Table 2 presents the thermal parameters of the investigated samples.

The thermal decomposition process of PEBSA_solution took place in two stages, as illustrated by the DTG curve. The first stage begins at 278 °C, reaches the maximum degradation rate at 338 °C, and records a total mass loss of 70.96%, this being the most pronounced degradation stage for this sample. This process is mainly attributed to the decomposition of squaric acid units, which leads to the release of CO, CO_2_, and H_2_O gases [34]. This temperature range also marks the beginning of the decomposition process of ethylene brassylate sequences [9]. In the second stage, an onset temperature was registered at 421 °C, and a maximum degradation rate at 454 °C, with a corresponding weight loss of 29%. This stage was assigned to the degradation of the hydrocarbon bonds in ethylene brassylate units, releasing fragments such as alkanes, carbonyl derivatives, and CO_2_. The thermal parameters recorded for PEBSA_solution have higher values than those belonging to comonomers and found in the literature, the data showing an improvement in the thermal properties of the copolymer. 

On the other hand, PEBSA_suspension has a slightly different thermal behaviour than PEBSA_solution. The differences are due to the synthesis conditions, namely, the presence of sodium dodecylbenzene sulphonate, which causes an increased molecular weight of the PEBSA_suspension samples. When comparing the TG curves of the two samples, it can be seen that PEBSA_solution has higher thermal stability up to 350 °C, whereas PEBSA_ suspension has greater thermal stability from 350 °C up to 675 °C. The main gases released during the decomposition process of PEBSA_suspension, which are analysed by mass spectrometry data and sustain the thermal behaviour of the copolymacrolactone system, are presented in the Supporting Information of the article.

The temperatures for mass losses of 10% (T_10_) and 20% (T_50_), respectively, confirm the previous statements. From the viewpoint of the residual mass, PEBSA_suspension has 9.78%, while PEBSA_solution is almost completely degraded, leaving only 0.04%. This difference is assigned to the presence of sodium dodecylbenzene sulphonate in the PEBSA_suspension samples. This surfactant exhibits a high thermal stability, while the decomposition process takes place in the range of 380 °C up to 480 °C, with a residual mass of approximately 40% at 700 °C [35]. The decomposition process of PEBSA_suspension shows four stages traced from the DTG curve. The first stage, with T_onset_ at 265 °C and T_peak_ at 286 °C, is assigned, as for the PEBSA_solution sample, to the decomposition of squaric acid units. The shifting of T_onset_ and T_peak_ to lower values can be ascribed to the presence of the surfactant molecules that induce larger inter-chain spacing, which affects the thermal stability [36]. The second stage of the decomposition process of PEBSA_suspension is attributed to the degradation of free surfactant, squaric acid, and to the beginning of the decomposition process of ethylene brassylate sequences. In the third and fourth stages, the most resistant structures are decomposed, namely, the ethylene brassylate sequences, a process similar to that encountered in the case of the PEBSA_solution sample, and the sodium dodecylbenzene sulphonate molecules. The specific thermal parameters of the analysed samples determined from DTA curves are presented in Table 3. 

PEBSA_suspension’s melting temperature and enthalpy variation values are higher than PEBSA_solution. The increase in the values of these parameters is ascribed to the higher molecular weight and rearrangements of the crystalline phase that occur in the copolymacrolactone chains synthesised by the suspension procedure.

In the following, the compound was characterised in terms of its formation, the availability of functional groups in the system, as well as cytotoxicity, and antioxidant character.

The initial goal when encapsulating erythritol in the PEBSA matrix was to obtain a compound with antioxidant properties as it is illustrated in Figure 5. At the same time, we considered new potential applications (for example, PEBSA can form gel-type structures—ongoing studies) of the new bioactive structure. In this context, we took into account studies in the field that mention Eryt as a compound that decreases the weight of dental plaque [23], has an inhibitory effect on the development of dual-species biofilm, or even induces the suppression of biofilm formation [37]; its presence decreases polysaccharide-mediated cell adherence, contributing to plaque accumulation [38], and can determine through a synergistic effect the inhibition of the growth of different cariogenic bacteria [39].

### 3.4. DLS Measurements

Dynamic light scattering (DLS), a sensitive, non-destructive method commonly employed for the characterisation of macromolecules in solution, was used to evaluate the ability of PEBSA_suspension and Eryt to form polymeric complexes. The parameters monitored by the DLS technique, the mean diameter, polydispersity index (PDI), and Zeta Potential (ZP) are summarised in Table 4. Figure 6 illustrates the size distribution of PEBSA_suspension, Eryt, and PEBSA_Eryt. 

A comparison of the size distribution shows a monomodal profile for all analysed samples. The mean diameter was found to be 596 nm for the PEBSA_suspension copolymer and 318 nm for Eryt. These results are consistent with previous results that have attested PEBSA’s ability to self-assemble [20]. The copolymer can adopt a globular shape as a result of the orientation of the carbonyl functional groups of the SA inside the spherical structures, a fact attested by the monomodal distribution. In the case of PEBSA_Eryt, a shift toward smaller sizes can be observed compared to the copolymer (with a peak at 487 nm), which can be attributed to the physical interactions between the copolymer and the polyol. The monomodal distribution together with this shift attests to the formation of the complex. At the same time, the PDI value for PEBSA_Eryt assembly decreases, indicating a relatively uniform assembly size. PDI values higher than 0.7 denote a broad particle size distribution, while lower values indicate a better homogeneity.

Finally, there were no significant differences between the ZP values for PEBSA_suspension (−25 mV) and PEBSA_Eryt (−24.9 mV) (Table 4). These data demonstrate that physical interactions take place between the initial compounds, leading to the formation of an assembly in which the polyol is embedded in the globular structure constituted by PEBSA_suspension. Moreover, the ZP values indicate good stability against coalescence.

### 3.5. Dynamic Vapour Sorption Behaviour 

As hydration and dehydration of systems influence their physico-chemical properties [40], the investigation of the dynamic vapour sorption behaviour of the PEBSA–Eryt compound constitutes an important analysis for future applications. The water vapour sorption capacity in the dynamic regime was evaluated (Figure 7). 

Taking into account the shape of the water sorption curves presented in Figure 7, they can be associated with isotherms of Type V, according to the IUPAC classification. This type of isotherm with hysteresis can be interpreted as being characteristic of porous surfaces and is the result of the complex interplay between the diffusion of water and the relaxation of the polymer matrix [41]. Moreover, such a type of sorption isotherms is specific for a hydrophobic material; the data are in good agreement with previous investigations concerning the hydrophilic/hydrophobic behaviour of PEBSA [21]. The compact structure of the copolymacrolactone generated by the supramolecular packing between the chains, including the physical bonds generated by the carbonyl groups of SA, but also the confirmation of the crystallinity of the polymer, results from the extremely reduced sorption/desorption process corresponding to PEBSA. The Eryt encapsulation in the PEBSA matrix determines a clearly different DVS behaviour for the PEBSA–Eryt compound, which is attributed to both the new pore structure and surface chemistry. It is assumed that the presence of Eryt as nano-fillers affects the water vapour permeability by modifying the polymer network, chain mobility, and crystallinity changes, which lead to the increasing access of the vapours inside the polymer system. These changes in the dynamic vapour sorption behaviour in polymer networks were also presented by other authors [42,43,44]. The changes in the crystalline morphology of the copolymacrolactone network due to the presence of Eryt are also confirmed by X-ray diffraction analysis. More than that, this aspect is sustained by the fact that squaric acid, comonomer in the PEBSA system, has the ability to crystallise if there is the possibility for interacting with hydrogen bond donors, in our case Eryt, data also presented in the literature [45].

Important information about the surface of the samples can be obtained from the dynamic vapour sorption investigation; thus, there is a reduced water vapour sorption at low values of relative humidity (RH), 0 to 10%, sometimes, a moderate sorption at intermediate values of RH, and a sharp increase in water sorption at RH values close to 100%. 

For determining the specific surface area (Table 5), the Brunauer–Emmett–Teller kinetic model (BET, Equation (2)) was implemented by simulating the sorption isotherms recorded under dynamical conditions
(2)$$W=\frac{W_{m}CRH}{\left(1-RH\right)\left(1-RH+CRH\right)}\;$$
where the control data involved are: weight of sorbed water—*W*, weight of water forming a monolayer—*W_m_*, sorption constant—*C*, and the relative humidity—*RH*.

The BET model characterises the sorption isotherms until a relative humidity of 40 %, depending on the sorption isotherm type and on the material type. This model can characterise type II isotherms, as well as types I, III, and IV. The dissimilarity between the order of water vapour sorption capacity and the values acquired for the specific surface may be induced by the nature of the functional groups of polymers. The sorption capacity of the studied samples can also be influenced by the average pore sizes in a complicated mode. 

Using the Barrett, Joyner, and Halenda model (BJF, Equations (3) and (4)), which is based on the determination methods for cylindrical pores, the average pore size, r_pm_, (Table 5) was established. The method applies the desorption branch of the isotherm. The desorbed quantity of vapour is caused either by the evaporation of the liquid core or by the desorption of a multilayer. The pore size distribution is specified as the distribution of pore volume. The relation between pore volume and the gas uptake can be explained if the density of the adsorbed phase is known. The first presumption of mesopore size analysis is that this phase is equal to the liquid phase of the adsorbed.
(3)$$V_{liq}=\frac{n}{100\rho_{\alpha }}$$
(4)$$r_{pm}=\frac{2V_{liq}}{A}$$
where *V_liq_* is the liquid volume, *n* is the absorption percentage, ρ_α_ is the adsorbed phase density, and A is the specific surface area evaluated by the BET method. The data presented in Table 5 confirm once again, by the moisture sorption capacity (*W*), the average pore size (*r_pm_*), and the specific surface area, the changes registered in the dynamic vapour sorption behaviour of the PEBSA–Eryt system. 

### 3.6. X-ray Diffraction Analysis

To obtain further evidence of the complexation ability, X-ray diffraction spectra were carried out and are depicted in Figure 8. The Eryt powder shows diffraction peaks at approximately 14.8°, 19.6°, 20.34°, 27.9°, 29.65°, 31.2°, and 41°, indicating a highly crystalline structure [46]. 

The diffraction patterns of PEBSA_suspension also show numerous diffraction peaks at 5.9°, 6.64°, 11.91°, 18.85°, and 23°. As expected for PEBSA_Eryt, its diffraction peaks are related to both the copolymer and the crystalline structure of the polyol, with no new characteristic peak appearing. This result demonstrates that the assemblies were formed by physical interactions. In addition, the sharp diffraction peaks indicate that the highly ordered structure of Eryt and PEBSA_suspension can be retained in the complex, also showing a good compatibility between components. 

### 3.7. Morphological Characterisation

In order to obtain a deeper perspective on the differences between the two copolymers, the morphology of PEBSA_solution and PEBSA_suspension was visualised by the SEM technique (Figure 9). Micrographs show that the polymers have a surface characterised by heterogeneity, with many protrusions of different sizes and shapes. However, it can be seen that, in the case of the PEBSA_solution copolymer, these protrusions are similar in appearance and more uniform, showing that the extent of heterogeneity of the samples depends on the polymerisation mechanism and on the reaction conditions. 

The morphology of the PEBSA–Eryt compound was investigated by TEM analysis. The images show that the assemblies have a roughly spherical shape (Figure 10) and their sizes ranged from around 5 ± 0.1 nm to 25 ± 0.2 nm. Even if the diameters of the polymeric assemblies measured by TEM were smaller than those found by DLS, the two characterisation methods demonstrate and support the encapsulation of Eryt in the PEBSA matrix. The differences appear due to fact that TEM measures the particle size in a dry state, while DLS determines the diameter of the particle together with the molecules or ions attached to its surface in solution (hydrated state) [47].

### 3.8. Antioxidant Behaviour of PEBSA–Eryt Bioactive Complex

DPPH has been widely used in the determination of the antioxidant activity of various compounds. The process relies on the reduction of alcoholic DPPH solutions in the presence of a hydrogen-donating antioxidant. The solution is transformed from a deep purple colour into the yellow product, diphenyl picryl hydrazine. Erythritol has proven antioxidant activity due to its hydroxyl groups, which function as good hydrogen donors [47]. Both samples exhibited significant antioxidant activity (Figure 11); the highest values of the scavenging activity percentage were observed, as expected, for Eryt (98.7%). However, PEBSA–Eryt also has a very high antioxidant activity of up to 86.2%, which proves that the antioxidant character of the polyol is preserved in the newly formed complex. Generally speaking, more hydroxyl means a higher electron transfer/hydrogen donating ability. Coupling Eryt with PEBSA led to new interactions between compounds, which resulted in a decrease in radical scavenging activity.

### 3.9. Biocompatibility of PEBSA–Eryt Bioactive Complex

The data regarding the use of erythritol in various in vitro and clinical studies highlighted the inhibitory effect on biofilm formation, as well as a positive cell-protective function provided by the polyol [24,48,49]. Furthermore, in vivo micronucleus tests demonstrated that erythritol is not mutagenic to bacterial cells and does not cause chromosomal damage in mammalian cells either in vitro or in vivo [50]. At the same time, aliphatic polyesters obtained by ring-opening polymerisation are used for bioapplications due to their biocompatibility [51,52].

According to the international standard for biological evaluation of medical devices (ISO 10993-5:2009), a reduction in cell viability by more than 30% is considered a cytotoxic effect. After 24 h of incubation with normal fibroblasts, as can be observed in Figure 12, the tested materials were found to have little effect on cell viability. Out of the four tested samples, it was found that cells treated with PEBSA–Eryt displayed a viability of around 91%, while the remaining samples did not significantly alter cell viability compared to the control.

## 4. Conclusions

The study confirmed the possibility to obtain poly(ethylene brassylate-co-squaric acid) in eco-friendly conditions, in aqueous suspension, free of harmful organic solvents. Thus, synthesised bio-based copolymacrolactone presented good thermal stability and higher molecular weights. Using the polymeric structure as a matrix for erythritol, a new bioactive system was prepared with antioxidant properties. FTIR and NMR analyses confirmed the structure of PEBSA and the new complex system and demonstrated the occurrence of physical intermolecular bonds between the synthesised copolymacrolactone and the polyol. DLS measurements attested to the ability of PEBSA to self-assemble, in which the copolymer takes on a globular shape as a result of the orientation of the carbonyl functional groups of the SA within the structures. Furthermore, the size distribution and ZP suggest that Eryt is embedded in the globular structure of PEBSA_suspension. The synergy between the polymer network and the antioxidant compound Eryt led to a structure with improved properties. Thus, Eryt encapsulation in the PEBSA matrix determines an increased dynamic vapour sorption capacity for the PEBSA–Eryt compound, attributed to the modification of the polymer network, chain mobility, and crystallinity. The changes in the crystalline morphology were also confirmed by X-ray diffraction analysis. DPPH radical scavenging activity assay revealed the antioxidant behaviour of the PEBSA–Eryt complex. Finally, both the PEBSA copolymer and PEBSA–Eryt assembly were proven to be cytocompatible, considering the in vitro cell viability responses of over 91% towards fibroblast cells. The research provides valuable information on the preparation of an effective and practical antioxidant structure with properties and potential for further development, particularly in the biomedical field. The bio-based nature of the compounds, as well as the eco-friendly conditions of preparation, recommend them for future use. 

## Figures and Tables

**Figure 1 antioxidants-11-02471-f001:**
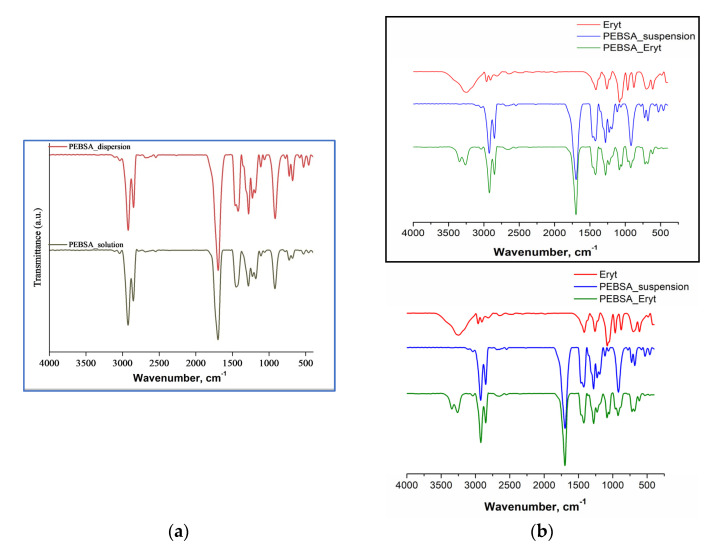
FTIR spectra of PEBSA_suspension and PEBSA_solution (**a**) and comparative PEBSA_Eryt samples (**b**).

**Figure 2 antioxidants-11-02471-f002:**
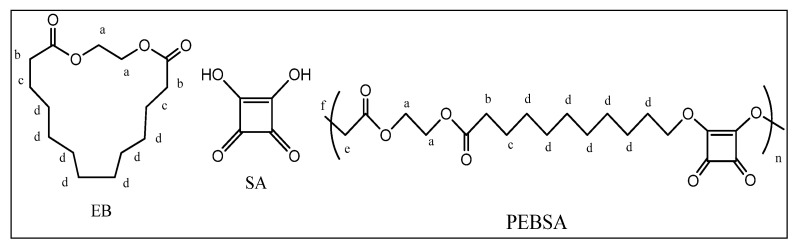
Chemical structure of monomers, copolymacrolactone, and notation of protons.

**Figure 3 antioxidants-11-02471-f003:**
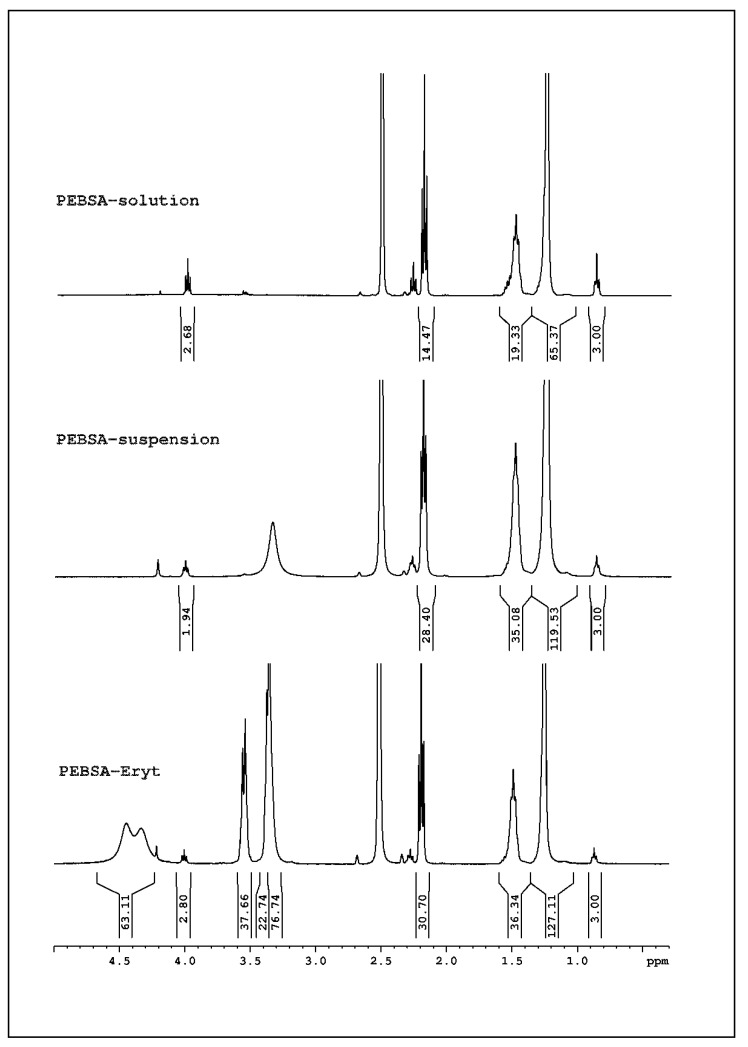
^1^H-NMR spectra of PEBSA_solution, PEBSA_suspension, and PEBSA_Eryt samples.

**Figure 4 antioxidants-11-02471-f004:**
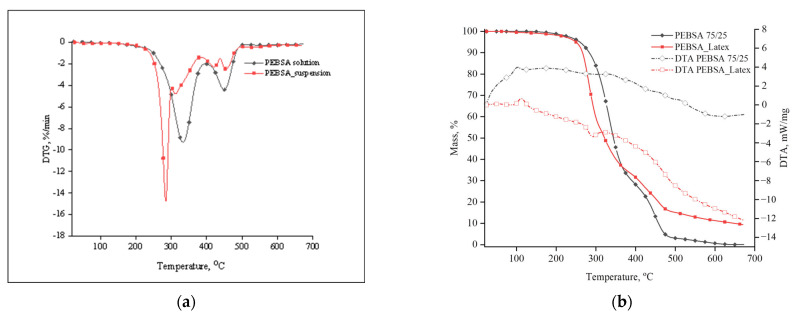
(**a**) DTG curve and (**b**) TG and DTA curve.

**Figure 5 antioxidants-11-02471-f005:**
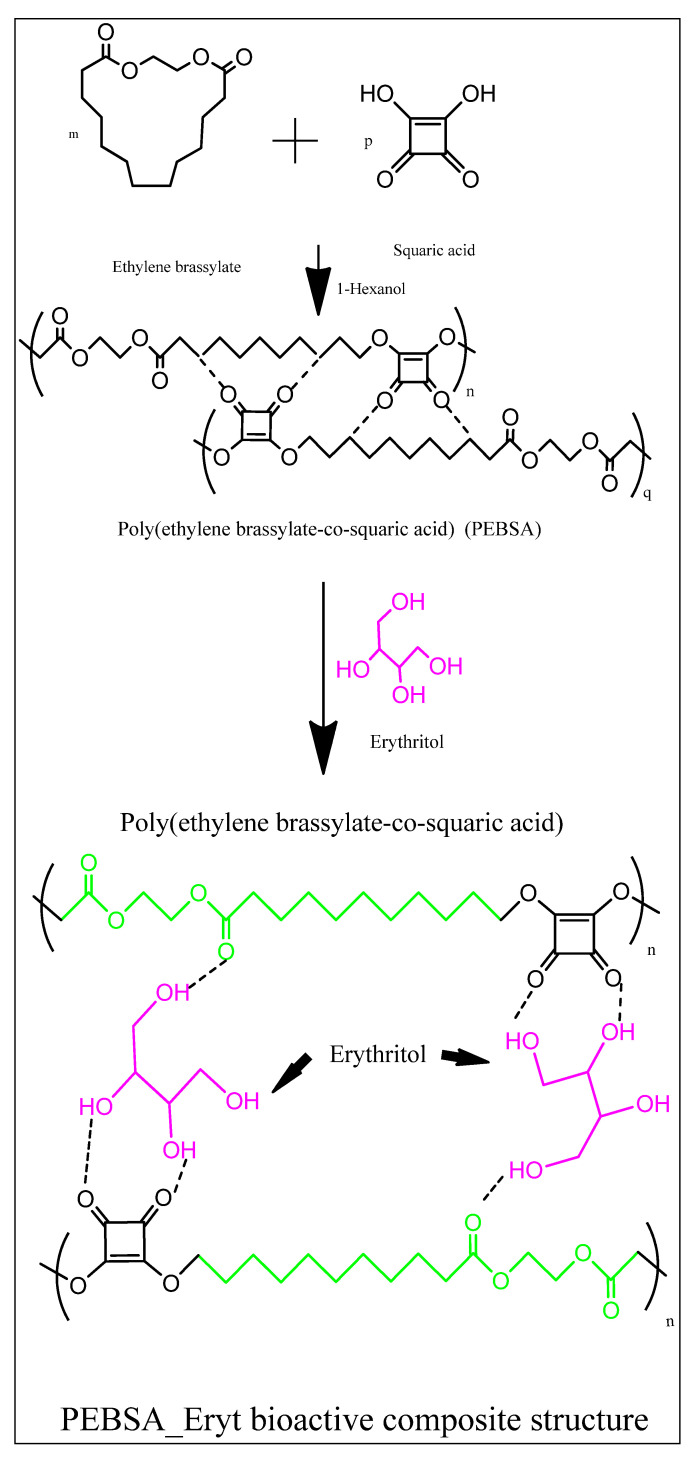
Schematised illustration of the synthesis of PEBSA and PEBSA_Eryt bioactive structure.

**Figure 6 antioxidants-11-02471-f006:**
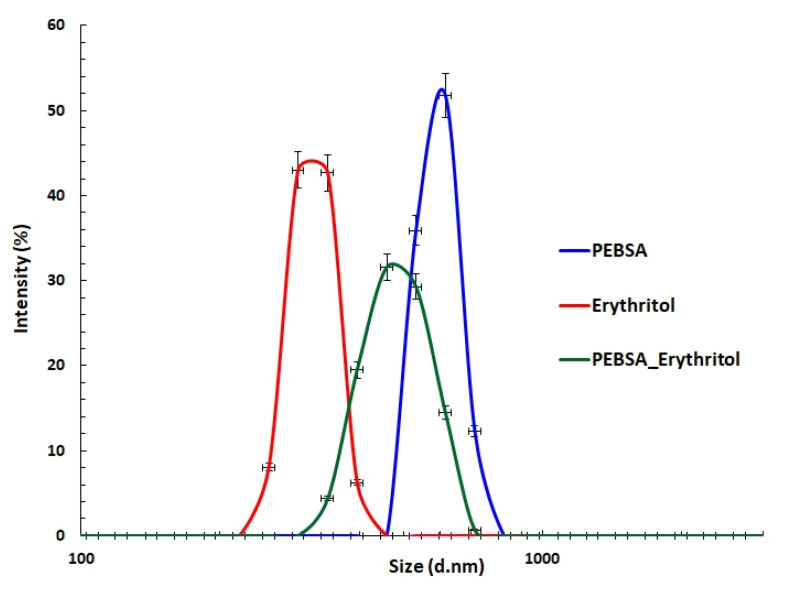
Size distribution of PEBSA_suspension, Eryt, and PEBSA_Eryt bioactive compound.

**Figure 7 antioxidants-11-02471-f007:**
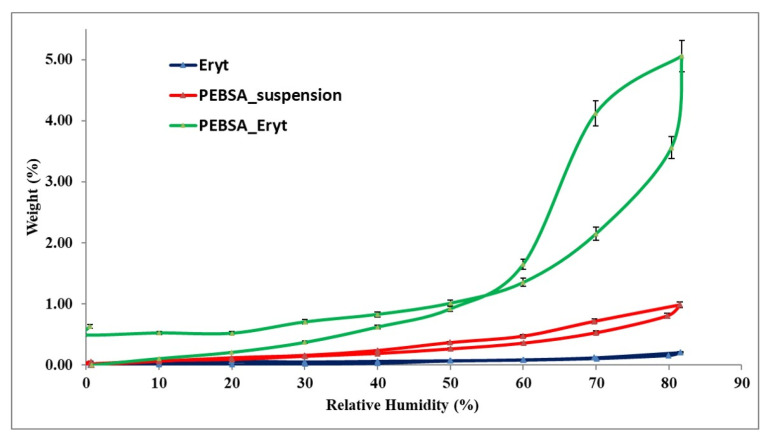
Sorption/desorption isotherms for the studied samples.

**Figure 8 antioxidants-11-02471-f008:**
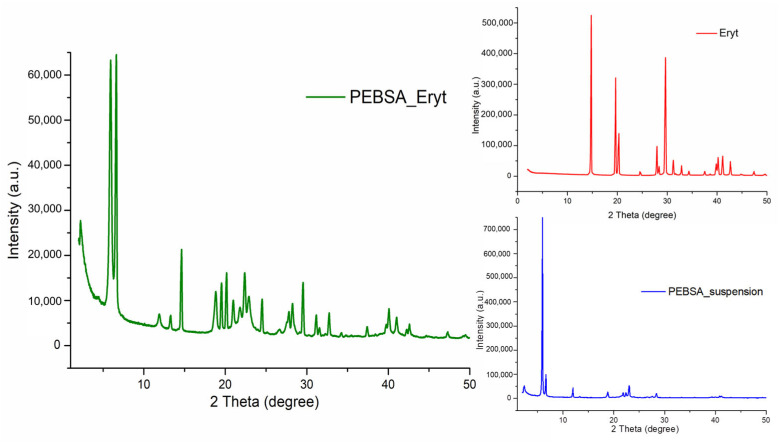
XRD diffractograms of PEBSA_Eryt, Eryt, and PEBSA_suspension.

**Figure 9 antioxidants-11-02471-f009:**
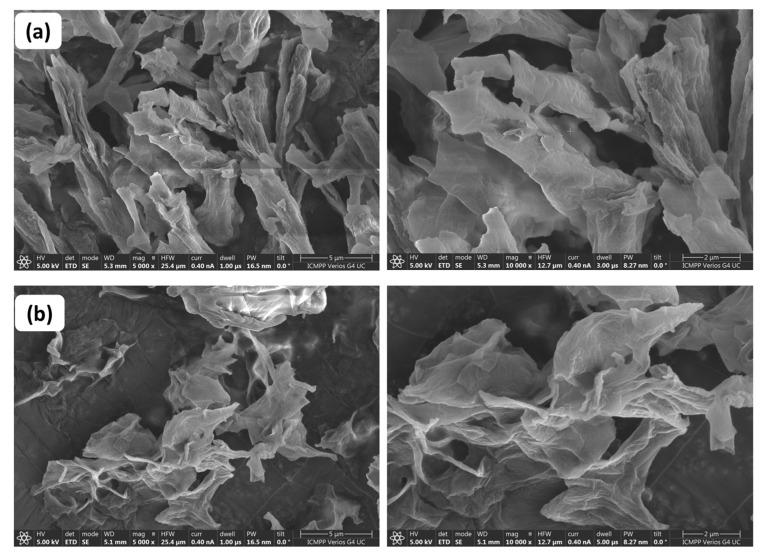
SEM images of the synthesised copolymers: (**a**) PEBSA_solution and (**b**) PEBSA_suspension at different amplifications (5 μm and 2 μm).

**Figure 10 antioxidants-11-02471-f010:**
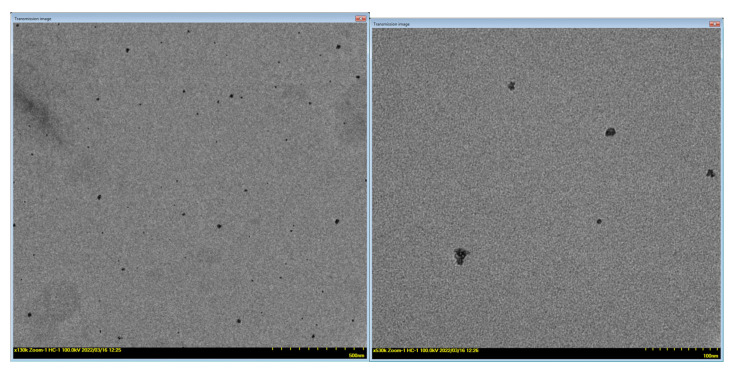
TEM micrographs of PEBSA–Eryt compound.

**Figure 11 antioxidants-11-02471-f011:**
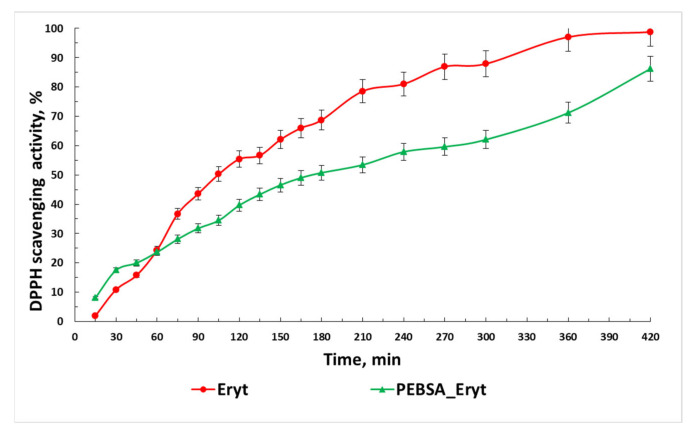
The scavenging activity percentage of Eryt and PEBSA_Eryt samples.

**Figure 12 antioxidants-11-02471-f012:**
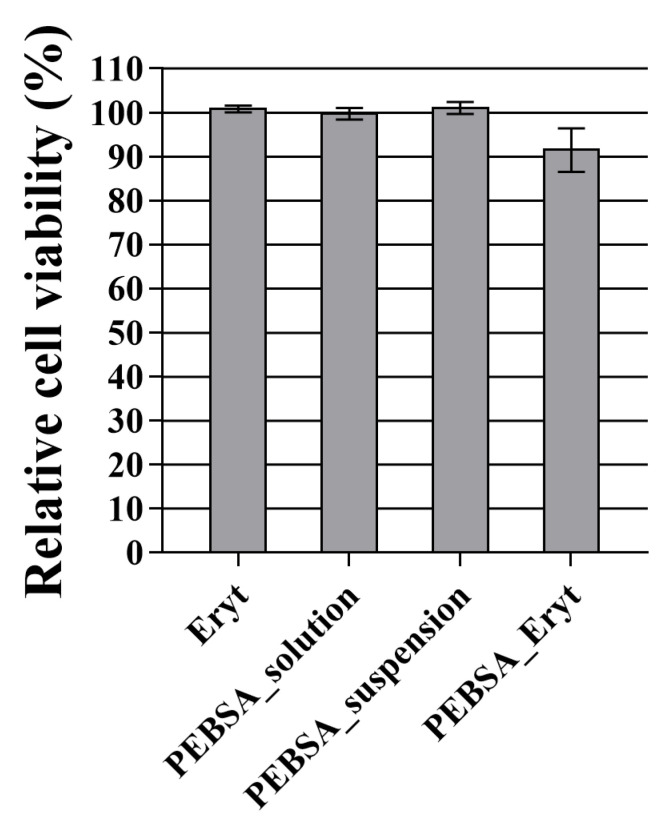
Cytotoxicity of HGF cells that were under treatment for 24 h with the tested compounds. The results are presented as a mean value ± the standard error of the mean (S.E.M.), *n* = 5; the differences were not statistically significant.

**Table 1 antioxidants-11-02471-t001:** Molecular weights of the synthesised copolymacrolactones.

Sample	M_p_	M_n_	M_w_	M_z_	M_z+1_	M_v_	D
PEBSA_solution	3785 ± 113	3977 ± 119	4006 ± 120	4076 ± 122	4137 ± 124	3992 ± 119	1.007
PEBSA_suspension	17,663 ± 529	22,703 ± 681	29,100 ± 873	42,848 ± 1285	62,605 ± 1878	27,081 ± 812	1.020

M_p_—Molecular weight of the highest peak; M_n_—Number average molecular weight; M_w_—Weight average molecular weight; M_z_—Higher average molecular weights; M_z+1_—Higher average molecular weights; M_v_—Viscosity Average Molecular Weight; D—Polydispersity index.

**Table 2 antioxidants-11-02471-t002:** Thermal parameters of the PEBSA samples.

Sample	Degradation Stage	T_onset_ °C	T_peak_ °C	W %	Residue	T_10_ °C	T_20_ °C
PEBSA_solution	I	278	338	70.96	0.04	287	310
	II	421	454	29			
PEBSA_suspension	I	265	286	40.29	9.78	268	283
	II	303	314	25.38			
	III	398	420	9.89			
	IV	442	456	14.66			

T_onset_—the temperature at which the thermal degradation starts; T_peak_—the temperature at which the degradation rate is maximum; T_10_, T_20_—the temperatures corresponding to 10% and 20% mass losses; W—mass losses.

**Table 3 antioxidants-11-02471-t003:** Differential thermal parameters.

	PEBSA_Solution	PEBSA_Suspension
T_m_ (°C)	104.4	114.1
ΔH (J/g)	71.36	86.33
ΔC_p_ (J/g·K)	0.819	0.522

T_m_—melting temperature, ΔH—enthalpy variation, ΔC_P_—heat capacity variation.

**Table 4 antioxidants-11-02471-t004:** Peak mean diameters and PDI values measured by DLS.

Sample	Peak (nm)	PDI	ZP (mV)
**PEBSA_solution ***	420 ± 12.6	0.9	
**PEBSA_suspension**	596 ± 17.8	0.52	−25 ± 0.75
**Eryt**	318 ± 9.5	0.67	−3.42 ± 0.1
**PEBSA_Eryt**	487 ± 14.6	0.49	−24.9 ± 0.74

* data presented in [20,21].

**Table 5 antioxidants-11-02471-t005:** Surface parameters evaluated based on adsorption/desorption isotherms: moisture sorption capacity; W, final weight; r_pm_, average pore size; and BET data.

Sample	W (%)	r_pm_ (nm)	BET Data *	
			Area (m^2^/g)	Monolayer (g/g)
**Erythritol**	0.2085	2.76	1.509	0.0004
**PEBSA**	0.9878	2.92	6.762	0.0019
**PEBSA_Eryt**	5.0588	3.54	28.575	0.0081

* Determined based on desorption branch of the isotherm (registered up to a relative humidity of 40%).

## Data Availability

Data is contained within the article and Appendix A.

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
