# Peer review of "Preparation of an Antioxidant Assembly Based on a Copolymacrolactone Structure and Erythritol following an Eco-Friendly Strategy"

_antioxidants, 2022, doi:10.3390/antiox11122471_

Round 1
Reviewer 1 Report
Aurica P. Chiriac et al, reported a novel assembly co-polymer structure. The eco-system is very unique for the community to do the interdisciplinary study and related bio-application. So the reviewer recommend it to be published in this journal under the minor revision.
1. The author should do a DFT simulation to under the mechanism between the co-polymer assembly (maybe use the monomer to do this for the convenient operation).
2. For the HNMR, the author should do a intensity-depandent study to avoid some missed information such as different concentration or temperature. Because that the intensity of NMR is related lot of a elements such as the concentration or temperature of your sample solution.
Author Response
Review Report_Reviewer 1
Comments and Suggestions for Authors
of the manuscript entitled:
Preparation of an antioxidant assembly based on a copolymacrolactone structure and erythritol following an eco-friendly strategy
authors
Aurica P. Chiriac, Alina Ghilan, Alexandru Serban, Ana-Maria Macsim, Alexandra Bargan, Florica Doroftei, Vlad Mihai Chiriac, Loredana Elena Nita, Alina Gabriela Rusu and Isabela Sandu
First of all, thanks to the reviewer for helping us improve our work. Corrections were made and the manuscript was rewritten according to the directions received.
Aurica P. Chiriac et al, reported a novel assembly co-polymer structure. The eco-system is very unique for the community to do the interdisciplinary study and related bio-application. So the reviewer recommend it to be published in this journal under the minor revision.
- The author should do a DFT simulation to under the mechanism between the co-polymer assembly (maybe use the monomer to do this for the convenient operation).
Thank you for the suggestion and recommendation; we are working on a study of combined molecular dynamics and a DFT simulation, especially due to the supramolecular structure that was highlighted by the X-ray investigation, and these will constitute the data of another article.
- For the HNMR, the author should do a intensity-depandent study to avoid some missed information such as different concentration or temperature. Because that the intensity of NMR is related lot of a elements such as the concentration or temperature of your sample solution.
Thank you for the suggestion and recommendation; concerning the 1H-NMR investigation of our samples in different conditions, namely concentration or temperature, we made the study and the recorded spectra at 23°C, 35°C, 45°C, 55°C and 65°C are illustrated in the following figures. From the superimposed spectra, no changes in the value of the integrals are observed. There are very small differences due to the processing of the spectra. The same behaviour was registered for different concentrations of samples.

Reviewer 2 Report
Abstract:
A good abstract should contain quantitative results. Such should be added.
Introduction: Add at the end ‘intention of the study’ and also write what is new or why this study brings value to the scientific community.
L 42-45: Pls. back that claim by providing values.
L 122: What %? wt.-%?
L 257-266: Belongs to ‘introduction’, ‘intention of the study’.
L 324-336: If the compounds change their structure without weight loss – that cannot be seen?
Author Response
Review Report_Reviewer 2
Comments and Suggestions for Authors
of the manuscript entitled:
Preparation of an antioxidant assembly based on a copolymacrolactone structure and erythritol following an eco-friendly strategy
authors
Aurica P. Chiriac, Alina Ghilan, Alexandru Serban, Ana-Maria Macsim, Alexandra Bargan, Florica Doroftei, Vlad Mihai Chiriac, Loredana Elena Nita, Alina Gabriela Rusu and Isabela Sandu
First of all, thanks to the reviewer for helping us improve our work. Corrections were made and the manuscript was rewritten according to the directions received.
- A good abstract should contain quantitative results. Such should be added.
The abstract was modified as it follows:
The study presents the achievement of a new assembly with antioxidant behaviour based on a copolymacrolactone structure that encapsulated erythritol (Eryt). Poly(ethylene brassylate-co-squaric acid) (PEBSA) was synthesised in environmentally friendly conditions, respectively through a process in suspension in water by opening the cycle of ethylene brassylate macrolactone followed by condensation with squaric acid. The compound synthesized in suspension was characterized by comparison with the polymer obtained by polymerization in solution. The investigations revealed that, with the exception of the molecular masses, the compounds generated by the two synthetic procedures present similar properties, including good thermal stability with Tpeak of 456°C, and capacity for network formation. Also, the investigation by dynamic light scattering techniques evidenced a mean diameter for PEBSA particles of around 596 nm, and a zeta potential of -25 mV, which attests to their stability. The bio-based copolymacrolactone was used as a matrix for erythritol encapsulation. The new PEBSA_Eryt compound presented an increased sorption/desorption process, compared with the PEBSA matrix, and a crystalline morphology also confirmed by X-ray diffraction analysis. The bioactive compound was also characterized in terms of its biocompatibility and antioxidant behavior.
- Introduction: Add at the end ‘intention of the study’ and also write what is new or why this study brings value to the scientific community.
The following paragraph was added to the introduction representing the intention of the study:
In this context, the study presents a more attractive alternative for the synthesis of the PEBSA copolymacrolactone system through an environmentally friendly approach that does not contain harmful organic solvents. By designing the synthesis of PEBSA in suspension, in the presence of a suitable dispersant, we aimed to obtain the new bio-based copolymacrolactone not only in eco-friendly conditions but also with improved characteristics. The comparative characterization of the copolymacrolactone synthesized by the two processes, in suspension and solution, was carried out, followed by the evaluation of the polymer structure as an erythritol coupling matrix with the aim of obtaining a new bioactive compound with antioxidant properties. Thus, the research provides information on the preparation of an efficient and practical biocompatible and antioxidant structure with properties and potential for further development.
- L 42-45: Pls. back that claim by providing values.
The phrase from L 42-45 was modified as it follows:
This is in the context where the synthesized homopolymer and other different EB-based block/copolymers provided an amphiphilic character, for example in the poly(ethylene glycol)-b-poly(ethylene brasylate) copolymer [10], a higher crystallization rate of about 67 °C and chains easily susceptible to hydrolytic degradation in the case of copolymers based on D, L-lactide and EB [13], improved mechanical behavior, flexibility and ductility compared to polylactides, since only a molar content of 5 % EB comonomer increases the elongation at break from 2 to 87 % [13] and also a self-assembled morphology, building block capacity and network formation.
- L 122: What %? wt.-%?
It represents the abbreviation for weight percentage, and it was corrected to 5 wt %.
- L 257-266: Belongs to ‘introduction’, ‘intention of the study’.
The phrase was improved and moved in Introduction.
- L 324-336: If the compounds change their structure without weight loss – that cannot be seen?
The structural changes during the thermal decomposition process were highlighted in the mass spectrometry data. Thus, the PEBSA_suspension main gases released during the pyrolysis process and presented in Supporting Information (the STA 449 F1 Jupiter thermobalance is coupled online with mass spectrometer Aëolos QMS 403C (Netzsch) as one system TG-MS) as MS spectra, extracted at 285°C, 429°C and 455°C temperatures at which the largest amount of gases released were recorded, are depicted from Gram Schmidt curves, while the identification of the volatile compounds was done using data from NIST MS library. In the text of the article, the following phrase was introduced to the thermal characterization:
The main gases released during the decomposition process of the PEBSA_suspension, which are analyzed by mass spectrometry data and sustain the thermal behavior of the copolymacrolactone system, are presented in the Supporting Information of the article.

Reviewer 3 Report
This manuscript entitled “Preparation of an antioxidant assembly based on a copolymacrolactone structure and erythritol following an eco-friendly strategy” is an interesting and original study.
The paper is clearly presented and results are very useful. However, I have some suggestions:
1. Line 256: Include the statistical analysis carried out.
2. Please include statistical results in tables (homogenous groups) and figures (error bars)
Statistical contrast is very important to be able to give certainty to scientific results.
Author Response
Review Report_Reviewer 3
Comments and Suggestions for Authors
of the manuscript entitled:
Preparation of an antioxidant assembly based on a copolymacrolactone structure and erythritol following an eco-friendly strategy
authors
Aurica P. Chiriac, Alina Ghilan, Alexandru Serban, Ana-Maria Macsim, Alexandra Bargan, Florica Doroftei, Vlad Mihai Chiriac, Loredana Elena Nita, Alina Gabriela Rusu and Isabela Sandu
First of all, thanks to the reviewer for helping us improve our work. Corrections were made and the manuscript was rewritten according to the directions received.
This manuscript entitled “Preparation of an antioxidant assembly based on a copolymacrolactone structure and erythritol following an eco-friendly strategy” is an interesting and original study.
The paper is clearly presented and results are very useful. However, I have some suggestions:
- Line 256: Include the statistical analysis carried out.
The following information concerning the statistical performed analysis was introduced in the article in Materials and Methods chapter:
2.4. Statistical analysis
The data presented in this study are the average of triplicate experiments, and have the standard error of the mean (SEM). Also, the statistical differences between data were made by One-Way ANOVA with Tukey test for the presented results. One-Way ANOVA with Tukey test and the bivariate Pearson Correlation were also used for the statistical differences between samples resulted for antimicrobial and biocompatibility tests of the samples.
- Please include statistical results in tables (homogenous groups) and figures (error bars)
Statistical contrast is very important to be able to give certainty to scientific results.
The statistical results were specified and included in the article in tables/figures and provide information on the analyzed characteristics.

Round 2
Reviewer 3 Report
Manuscript has been improved considerably.